# How is Satiety Affected When Consuming Food While Working on A Computer?

**DOI:** 10.3390/nu11071545

**Published:** 2019-07-08

**Authors:** Feng Ding, Nazimah Hamid, Daniel Shepherd, Kevin Kantono

**Affiliations:** 1Department of Food Science, Auckland University of Technology, Private Bag 92006, Auckland 1142, New Zealand; 2Department of Psychology, Auckland University of Technology, Private Bag 92006, Auckland 1142, New Zealand

**Keywords:** satiety, eating behavior, meal intake, stress, eating traits

## Abstract

More people working at offices are choosing to eat meals at their desks, making “desktop dining” an increasingly common phenomenon. Previous studies have reported that environmental distractors, such as television viewing, can influence meal intake and subsequent snack intake. However, the impact of stressful mental tasks on eating behavior has received relatively less attention, focusing only on subsequent meal intake or concurrent snack intake. This study sets out to determine whether eating while working influenced current meal energy intake. This research also examined the relationship between dietary restraint status and energy intake. A crossover experimental design was employed requiring participants (14 males and 29 females) to eat pizza quietly and at rest (control), and while working on a computer (work). Measurements included BMI, energy intake, state anxiety, restrained eating behavior, stress levels (pre- and post-eating), and appetite (before and after both work and control sessions). The findings showed that consuming food while working on a computer significantly increased stress but had no influence on energy intake compared to the control condition. However, post-eating hunger levels were significantly higher in the work condition compared to the control condition. As expected, satiety levels decreased significantly from pre- to post-eating for both work and control conditions. In addition, no significant relationship was observed between restrained eating behavior and energy intake in both work and control conditions. These results suggest that eating while working affected satiety of normal weight participants, as indicated by the significant difference in post-meal satiety levels between work and control conditions.

## 1. Introduction

Globally, cases of obesity are increasing at an alarming rate. The World Health Organization reported that 39% of adults were overweight in 2016, and 13% were obese [1]. Over the past few decades, researchers have been seeking reasons and solutions for this health problem, including the biological control of appetite. Satiety has been reported to be impacted by internal physiological factors, such as appetite [2], as well as external factors, such as environmental stimuli, such as television viewing [2,3,4,5,6,7]. As an extremely hungry state is rarely experienced in developed countries, eating behavior depends heavily on the response of people to external factors rather than the basic homeostasis physiological function of the human body [8]. 

Numerous studies have attempted to document how energy intake can be influenced by environmental stimuli, such as visual and auditory distractors. Hetherington et al. [2] and Bellisle et al. [4] concluded that consuming food while viewing television resulted in significantly higher energy intake than eating quietly alone. Blass et al. [2] further stated that significantly higher energy intake was noted in the television viewing group compared to the music listening group. Bellisle and Dalix [9] reported that eating while listening to a story resulted in significantly higher energy intake than eating alone. 

In addition to energy intake, satiety can also be measured subjectively using self-reported appetite ratings [10], which can be influenced by physical, physiological, and psychological factors, before, during, and after a meal [11]. The results of appetite rating after meals can vary due to the influence of different eating conditions. Playing computer games while having lunch resulted in significantly less post-lunch fullness than eating lunch with no distractor [12,13]. Bellisle et al. [5] further reported that eating with two other people resulted in significantly lower post-meal fullness than when eating alone, watching television without food cues, watching television with food advertisements, and when listening to a detective story. Born et al. [14] found that participants attempting an unsolvable mathematical test had significantly lower satiety scores than those doing solvable mathematical test.

In addition to the different effects of various environmental stimulus on appetite ratings, the biological demands of mental work differ from physical work. Thus, it is necessary to figure out the impact of cognitive tasks on appetite sensation and energy intake. Mental work refers to a task needing cognitive efforts [15], and food intake can be related to stress levels brought about by cognitive tasks. To investigate the relationship between psychological stress and energy intake, studies in laboratory settings have used different mental tasks to create acute stress. However, not all research which explored the association between food intake and task-induced mental stress have yielded conclusive results. In contrast to Born et al. [14], Rutters et al. [16] reported that participants doing an unsolvable math test had significantly higher energy intakes than those who did a solvable math test. Chaput and Tremblay [15] and Chaput et al. [17] both reported that energy intake was significantly higher in participants undertaking a reading–writing task compared to those who did nothing. However, Salama et al. [18] found that male participants doing a reading–writing task had significantly lower energy intakes compared to participants at rest.

Dietary restraint is increasingly important for the control of eating behaviors and is a synonym for chronic dieting, which is a state of trying to restrict food intake [19]. Working on a task that distracts people may weaken their self-monitoring of food and influence food intake. However, this restraint depends on the extent to which individuals are dietary restrained [9]. Zellner et al. [20] reported that restrained eaters performing stress-inducing mental tasks consumed more than those in the no-stress group. Royal and Kurtz [21], on the other hand, found no significant interaction between dietary restraint and energy intake in both low-stress and high-stress participants working on solvable and unsolvable anagrams, respectively. 

Eating while working is increasingly becoming common in modern society. There is currently no study estimating the impact of eating while working on concurrent food intake. Therefore, the current research sought to determine the impact of computer work on energy intake in a controlled experiment. This research aimed to determine if a work-related stressor can influence satiety. In two experimental conditions, one with and one without a work task, measures of hunger, fullness, and food pleasantness will be compared. In addition, the influence of dietary restraint upon eating while working in terms of energy intake will be investigated. 

## 2. Materials and Methods

### 2.1. Participants

Forty-three participants were recruited through poster advertisements, and the Auckland University of Technology Ethics Committee (AUTEC) provided ethical approval (EA 18/179) for this study. All participants gave written informed consent before the commencement of the study. Inclusion criteria included being aged between 18 and 65 years old, being in good health, and consuming food regularly while working. Participants having food allergies (such as gluten, milk, eggs, wheat, ham, and food preservatives), who are on a vegan, vegetarian or kosher diet, or who have medical conditions or take medication which can affect appetite, were excluded from this study. Before the experiment, participants were asked to eat breakfast and fast for two hours, ensuring that they had sufficient appetite to complete the experiment. 

### 2.2. Experimental Design

A crossover design was applied, incorporating a mental work condition and a control condition. The experiment was conducted between 11 am and 12 pm on weekdays in a custom-built sensory laboratory. Figure 1 summarizes the research program. In the control condition, participants were required to consume food quietly while in isolation. In the working condition, participants were required to perform a computer-based task while consuming a midday meal. The task involved copying columns of numbers from data files into a master file over a 20-minute period, all while consuming pizza. Forty participants attended both sessions, with the order randomized. Forty participants were chosen to carry out the experiments in conditions to achieve a statistical power of 0.90 to 0.95 using Cohen’s d calculation of 0.8 [22]. For each participant, the control and working conditions were carried out separately with a 1-week interval, but at the same time and day of the week.

### 2.3. Food stimulus

Ham and cheese pizza (Dominos, New Zealand) was provided as a lunch meal. One pizza provided 5848 kJ and contained 52 grams (g) protein, 28 g fat, 223.2 g carbohydrate, 9.6 g sugars, and 3.368 g sodium (https://www.dominos.co.nz/media/3550/nz-value-range-2017.pdf). All participants were provided with the same pizza. They were given 20 minutes to consume the pizza and were permitted to drink an unlimited amount of water during the experiment. 

### 2.4. Experimental Procedure

On the day of experiment, participants were required to consume breakfast and then asked to consume nothing else besides water. Upon arrival at the sensory laboratory, the weight, height, and waist circumference of participants were measured. Next, participants completed a Dutch Eating Behavior Questionnaire (DEBQ; [23]) to examine if they were restrained, emotional, and external eaters. The current state of anxiety was then assessed using the state anxiety dimension from the State–Trait Anxiety Inventory questionnaire (STAI; [24]). The total scores of STAI ranged from 11 to 52, with higher scores indicating greater state anxiety. The single-item stress scale [25] was used to assess participants’ current stress level. This scale produced a range of scores from 0 to 10. The higher the scores, the greater the reported stress. Moreover, the Visual Analogue Scales for hunger and fullness were employed to evaluate participants’ satiety level. Next, a whole pizza was presented, and participants were required to evaluate its pleasantness and palatability before eating it. Participants then had 20 minutes to consume the pizza, with energy intake determined by the weight of pizza consumed. 

After eating the pizza, the participants’ state of stress and satiety were measured again. Finally, a new pizza was presented to participants for them to assess pleasantness and palatability again, but this time, they were not allowed to eat it. Each step in the two experimental conditions was the same except that participants were either given a task while eating or else ate without any distraction.

### 2.5. Measurements

#### 2.5.1. Demographic (BMI and Waist Circumference)

Each participant’s body mass index (BMI, kg/m^2^) and waist circumference were calculated. BMI was determined using the participant’s weight and height. Participants were asked to take off their heavy outerwear, such as jackets and coats, and shoes. Height and waist circumference were measured by a flexible, but non-stretchable measuring tape. Weight was measured by a scale (Living & Co, Auckland, New Zealand) placed on the ground. It was essential to determine BMI and waist circumference of participants because the physiological and behavioral responses to satiety, appetite and hunger between lean, over-weight, and obese participants are likely to be different [26].

#### 2.5.2. Dutch Eating Behavior Questionnaire

The Dutch Eating Behavior Questionnaire (DEBQ; [23]) was employed to evaluate whether participants were restrained, emotional, or external eaters. This self-reported evaluation tool was composed of three different subscales containing 33 items. The Emotional Eating scale had 13 items, while the External Eating scale and Restraint Eating scale contained 10 items each. Each item in the DEBQ was rated with a 5-point Likert scale anchored with “never” to “very often” at each end. van Strien et al. [27] noted that the mean score of the DEBQ Restrained Eaters scale was referred to as the restrained score. Participants with restrained scores greater-or-equal to three were restrained eaters, while those with restrained scores less than three were non-restricted eaters. The results obtained by administering this questionnaire are useful when explaining differences in food intake [28,29]. 

#### 2.5.3. State–Trait Anxiety Inventory Questionnaire

The State–Trait Anxiety Inventory (STAI) questionnaire comprises 20 questions assessed using a 4-point Likert scale that ranged from 1 (not at all) to 4 (very much so) [24]. Spielberger et al. [24] defined state anxiety as "the conscious and subjective feeling of discomfort and tension, which was accompanied or associated with the level of arousal or arousal caused by the autonomic nervous system". Martens et al. [30] argued that state anxiety occurs when individuals consider the requirement of an objective situation as being a threat, which can be attributed to the inability of individuals to perceive themselves as meeting the requirement of an objective environment. Hence, this study adopted the STAI-state questionnaire to assess the current anxiety state of participants. 

#### 2.5.4. Single Item Stress Scale

A previous study by Karvounides et al. [25] showed that the Stress Numerical Rating Scale-11 (SNRS-11) can be efficiently used to assess current stress levels in adolescents and adults. In this study, participants completed the SNRS-11 single-item stress scale to evaluate their degree of stress. The stress scale was used to determine stress when participants felt tense, restless, or overwhelmed. On a scale of 0 to 10, with 0 being “no stress” and 10 being “worst stress” possible, participants indicated their current level of stress. We hypothesized that participants might be stressed when completing tasks when eating and that the changes in stress before and after eating pizza under the two different consumption conditions might help explain the differences in satiety and food pleasantness, if any. 

#### 2.5.5. Satiety and Food Pleasantness

The use of visual analog scales (VAS) is a valid approach to measure feelings, such as satiety and pleasantness during food consumption [31], and has good repeatability between groups [32]. The changes in satiety and food pleasantness during food consumption were recorded using a 100 mm VAS anchored at each end with words describing the poles of a unidimensional question. The ten VAS items were scored both before and after the meal. The questions on satiety and food pleasantness are presented in Table 1 and Table 2.

#### 2.5.6. Food Intake

Participants were required to eat until they felt satiated. The pizza was weighed before and after each meal, and food intake was determined by the weight difference. Energy intake (kJ) was calculated and translated based on the weight of pizza consumed (g) using the nutritional value information provided on the New Zealand Dominos website (https://www.dominos.co.nz/media/3550/nz-value-range-2017.pdf).

### 2.6. Statistical Analysis

For state anxiety, the scores of the twenty STAI questions were summed to yield a total score. For the DEBQ, average values were taken for each eating behavior dimension (i.e., restraint, emotional, and external). For the satiety and food pleasantness visual analog scales, a total sum score of hunger was calculated by summing up the scores for questions on hunger, fullness (reversed scoring), desire to eat, predicted consumption, and satiation. Pre- and post-average food pleasantness were obtained by averaging the scores for items on expected appearance, odor, pleasantness, taste, and enjoyment. To test whether the different conditions induced stress and whether there was an influence of working condition on energy intake, a two-way Analysis of Variance (ANOVA) was carried out that compared pre- and post-conditions for both control and work conditions on stress and energy intake ratings. An additional one-way ANOVA was then carried out to further evaluate the differences between pre- and post-ratings for each condition for stress and energy intake. 

Analysis of covariance (ANCOVA) was carried out to explore how the dependent variables of energy intake, hunger ratings, and food pleasantness were influenced in the control and work conditions, while statistically controlling for the effects of other continuous variables, which are not of primary interest, such as physical characteristics (BMI, waist circumference, age, and gender) and psychological traits (DEBQ and STAI scores) that are known as covariates. Additionally, multiple linear regression analysis and ANOVA were carried out to further investigate the influence of age and gender, respectively. Partial Least Squares Path Modeling (PLS-PM) analysis was also carried out to further summarize the relationship between physiological, psychological, and stress measures on satiety, food pleasantness, and energy intake. Model reliability was tested using Goodness of Fit (GoF) statistics, Cronbach’s α, Dillon–Goldstein’s *rho*, and *R*^2^. The GoF also assessed the overall predictive performance of the model by considering the communality and the *R*^2^ coefficients. All univariate and multivariate statistical analyses were performed using R v 3.5.1 and RStudio v 1.1.463. R packages that including FactoMineR (v 1.41) and PLS-PM (v 0.4.9) were used.

## 3. Results

All 43 participants completed their experiments successfully and strictly adhered to the study protocol. The analysis was performed on all data collected before and after each meal.

### 3.1. Participant Characteristics

Participants were 25.79 ± 4.87 years of age. Participants had average BMI and waist circumference of 21.75 ± 2.75 kg/m^2^ and 76.08 ± 7.25 cm, respectively. On average, the DEBQ scores for dietary restraint, emotional eating, and external eating score were 2.46 ± 0.77, 2.49 ± 1.04, and 3.28 ± 0.49 respectively. The STAI score was 26.09 ± 8.53. All data are reported as mean ± standard deviation (see Table 3).

### 3.2. Work Induces Stress 

Overall, a significant increase in stress was observed for the work condition compared to control condition (*F*_(3,171)_ = 61.98; *p* < 0.001) as seen in Figure 2. Interestingly, further analysis showed that the post-control condition showed significantly lower stress ratings compared to the pre-control condition (*F*_(1,85)_ = 19.78; *p* < 0.001). As expected, the work condition resulted in significantly higher stress (*F*_(1,85)_ = 58.36; *p* < 0.001) post-work compared to pre-work. These findings indicate that the assigned task in the work condition was sufficiently demanding. 

### 3.3. Overall Influence of Physical and Psychological Characteristics 

ANCOVA revealed significant differences for energy intake (*F*_(10,85)_ = 2.78; *p* = 0.006) and post-eating hunger (*F*_(10,85)_ = 6.25; *p* < 0.001) across control and work conditions, but not for post-eating food pleasantness ratings (F_(10,85)_ = 1.63; *p* = 0.112). For energy intake, a significant effect was observed for age (*F*_(10,85)_ = 6.89, t_age_ = −2.625; *p* = 0.011) and gender (*F*_(10,85)_ = 6.36, t_female_ = −2.522; *p* = 0.014) only. For post-eating hunger ratings, marginal significance were observed for stress (*F*_(10,85)_ = 3.94, t_stress_ = 1.985; *p* = 0.051) and condition (*F*_(10,85)_ = 3.94 t_control_ = −1.937; *p* = 0.057). Interestingly, psychological characteristics, such as DEBQ dimensions (restraint, emotional, and external) and STAI, did not influence energy intake, hunger, and food pleasantness ratings.

Significant differences were observed for energy intake between male and female participants in both the control (*F*_(1,42)_ = 4.99, t_female_ = −2.236; *p* = 0.03) and work (*F*_(1,42)_ = 5.05, t_female_ = −2.247; *p* < 0.05) conditions (Figure 3). Overall, age was shown to be a significant predictor (*t*_(170)_ = −2.71; *p* = 0.008) for energy intake, but with no significant differences found between the work and control conditions. A significantly higher post-eating hunger rating was found in the work condition (x_work_ = 19.3 ± 2.22) compared to control (x_control_ = 13.79 ± 4.42) (*F*_(1,85)_ = 53.35; t_control_ = −7.305, *p* < 0.001). Additionally, stress was found to induce significantly higher hunger ratings (*t*_(170)_ = 7.51; *p* < 0.001) in the work condition. Table 4 provides an overview of the descriptive results of the measured variable in this study.

### 3.4. Partial Least Square Path Modeling 

To illustrate the relationship between physical (BMI, Waist circumference, Age, and Gender) and psychological (self-rated stress and eating behavior) measures, as well as energy intake, hunger, and food pleasantness, PLS-PM analysis was carried out (Figure 4). Cronbach’s α, Dillon–Goldstein’s *rho*, and *R*^2^ values of each variable were 0.822, 0.854, and 0.836, respectively, indicating overall reliability of the measured variables. The GoF measure was similar to the GoF of the bootstrapped model (GoF = 0.549). Interestingly, stress was shown not to be correlated with either psychological (DEBQ and STAI) or physical traits measured in this study. A significant positive correlation, however, was shown between stress and energy intake, while a significant negative correlation was shown for stress and food pleasantness measures. Further internal modeling was also carried out to explore the relationship between the response variables in relation to energy intake. Although hunger showed a positive relationship with energy intake, it did not reach significance. However, food pleasantness showed a significant negative correlation to energy intake.

## 4. Discussion

### 4.1. Eating while Working Influenced Stress

The current study showed changes in stress levels before and after eating pizza under control and working conditions. The results showed that when participants were required to complete tasks on a computer while eating pizza for a maximum period of twenty minutes, they experienced significantly greater stress compared to control condition (re Section 3.1). This finding illustrated that performing a computer task could effectively increase stress, as confirmed in previous studies. Studies have reported that various stressors, including anagram [20,21], reading-writing task [15,17,18], mathematical test [14] and mental arithmetic task [16], significantly increased stress. 

### 4.2. Energy Intake in Work and Control Conditions Was the Same

It was hypothesized in this study that energy intake in the work group would be higher than the control group due to stress-induced eating. However, interestingly, no significant difference in energy intake was found between the work and control conditions with the normal weight participants used in our study. The fact that there was no significant increase in energy intake under stressful working conditions was in keeping with results reported by Lemmens et al. [36], who showed no significant difference in terms of energy intake between normal weight participants who did a math test and those who were resting, although those doing the math test felt significantly more stressed than those at rest. However, in their study, visceral overweight participants under the stress condition had significantly higher energy intake than those under the rest condition [36]. Similarly, Rutters et al. [16] who estimated energy intake of participants carrying out a solvable and an unsolvable maths problem in the control and stress conditions, demonstrated that overweight participants had significantly higher energy intake of sweet snack food in the stress group compared to the control group. Recently, Siervo et al. [37], who recruited obese male participants, found that participants playing a violent video game experienced significantly more stress and had significantly higher energy intake than participants watching non-violent television. 

All the studies above suggest that food intake was more likely to increase for obese individuals when they are under stressful conditions compared to normal weight individuals. Hence, acute psychological stress did not affect energy intake of normal weight participants, as confirmed in this study. It has been found that under stress-free conditions, the homeostatic system controls eating to maintain energy balance, while the reward system increases intake of energy-dense and palatable foods under stress conditions in visceral overweight participants when compared to normal weight participants [20,36]. Thus, although stress usually drives people to consume more food, normal weight people appear to control how much they eat. The average body mass index of participants in the current study was 21.75 ± 2.75 kg/m^2^, which was within the normal range of 18 to 25. This finding supports the notion that successful management of body weight involves a complicated interplay of both situational factors and internal traits [21]. Hence, it would be interesting to carry out further studies on the impact of work-related stress on food intake with obese participants to investigate if significant changes in energy intake between control and work conditions occur.

### 4.3. Stress Drives an Increase on Energy Intake

The differences in energy intake between control and work conditions were not significant. However, further analysis using PLS-PM showed that stress was a main driver for increased energy intake (re Section 3.3). Some studies have also reported that normal weight participants with induced stress ate significantly more food than those who were not stressed. Chaput and Tremblay [15] reported that participants who had completed reading–writing tasks had significantly higher energy intake than those at rest. Later, Chaput et al. [17] replicated this result but found no difference in energy intake between two different working conditions: reading–writing and a computerized test despite both the studies involving participants consuming food post-task. These contradicting findings can be explained by the cognitive task being either a within-meal distraction or independent distraction before the meal. In both studies conducted by Chaput and colleagues [15,17], participants were required to complete cognitive tasks before food consumption, while in the present study, participants performed the mental task while eating. Ogden et al. [7] explained that people might forget to eat when they engage in a cognitive task. In fact, the behavior related to food consumption requires some cognitive effort, and as distraction increases, participants have less cognitive ability when eating. Participants in the current study were urged to do as much of the tasks as possible while eating, which might greatly distract them from eating. However, in the studies conducted by Chaput and Tremblay [15] and Chaput et al. [17], food was provided after performing the cognitive tasks, which might not influence participants’ cognition while eating. Furthermore, it is important to note that both studies recruited a small sample size of participants (*n* = 15 and *n* = 17, respectively) and as from a statistical viewpoint was underpowered. 

Changes in snacking behavior have been shown to increase under stressful conditions. Royal and Kurtz [21] and Zellner et al. [20] explored the effects of mental work on snack food consumption while doing solvable and unsolvable anagrams. Royal and Kurtz [21] pointed out that participants experiencing more stress had significantly higher energy intake of snacks than those with less stress. Moreover, Zellner et al. [20] reported that participants consumed significantly more unhealthy food under stressful conditions, such as chocolate, and less healthy food, such as fruits. The food used in both these studies were afternoon snacks rather than a substantive meal, while the present study required participants to eat lunch. Hence, underlying reasons on how food consumption when performing a task influences energy intake has yet to be determined.

Generally, it is known that when people do high-level cognitive tasks, their ability to monitor how much they consume is impaired, leading to increased food intake than those who do not complete cognitive tasks [3,9,13]. A possible explanation as to why the stress condition in the current study may not have been sufficient to trigger stress-eating is the fact that people do not eat more at a sitting but rather eat more often. Thus, snacking behavior of participants who complete mental tasks should be further investigated rather than just focusing on meal intake while performing mental tasks. 

### 4.4. Gender Variation on Energy Intake

The differences in energy intake between genders were also examined in this study. There was no difference between the control and work groups for either male or female participants in terms of energy intake. Similarly, Rutters et al. [16] concluded that gender did not significantly affect energy intake after doing solvable and unsolvable mental arithmetic tasks. In contrast, Salama et al. [18] reported that females had significantly higher energy intake after completing a reading–writing task compared to those who had a rest, while energy intake for males tended to be lower, with a significant decrease in dessert consumption, after the reading–writing task. The authors concluded that women reacted differently after mental work and explained that when exposed to stressors, females tend to eat food to escape the stressed feeling, while males preferred to consume alcohol and tobacco [38]. However, Torres and Nowson [38] specifically stated that food intake was only significantly related to stress in obese women. 

In addition, energy intake in both control and work conditions were significantly higher for male participants compared to female participants in the current study. This finding was in line with Hetherington et al. [3] and Salama et al. [18], who also found that in all the conditions, males had significantly more energy intake than females. This difference in gender may be explained by psychosocial mechanisms, whereby women tend to eat less to appear more feminine, while men perceive eating as a sign of power and masculinity [39]. 

### 4.5. Hunger Level Was Higher in the Work Condition

Hunger scores did not significantly differ between the work and control groups before the consumption of pizza. This result supports a state of equal satiety amongst participants, which is in line with all previous research [3,4,5,7,13,14,15,16,40,41]. Ensuring that participants’ appetite was at the same level between the work and control conditions before food consumption is necessary to measure subsequent energy intake and satiety after consuming food.

The second hypothesis in this research was that satiety would differ after consuming pizza between work and control groups. In this study, the work group had a significantly higher post-meal hunger level than the control group (re Section 3.2), which supported our hypothesis. This finding was similar to the study carried out by Born et al. [14], who pointed out that participants doing an unsolvable mathematical test had significantly less post-meal satiety scores than those doing a solvable mathematical test. Moreover, no difference in energy intake was detected between participants in these two conditions. Hence, results in the current study and Born et al. [14] inferred that people who were stressed had similar food intake but were significantly less full after eating compared to those who were not stressed. It has been hypothesized that increased post-meal appetite following a stressful cognitive task may be due to impaired satiety signals or stress-induced rewards [42]. Chaput et al. [17] also concluded that demanding cognitive tasks have the potential to change satiety signals and can exacerbate the disorder between food consumption and appetite sensation. This further indicated that psychological factors related to stress were strongly correlated with the urge to eat compared to the actual amount eaten. This finding supported the notion that subjective hunger motivation could be enhanced by emotional stress [17], and perceived stress was related to greater hunger [43]. Another potential explanation for the post-meal difference in the fullness of participants between mental work and non-work conditions was that distraction limited the capacity of participants to pay attention to visceral sensations produced by food consumption, which may result in less fullness [12].

Some studies showed no significant difference in post-meal satiety between mental work and no work conditions. Chaput and Tremblay [15] pointed out that no significant difference in hunger levels after eating between participants who completed the reading–writing task and those who had a rest. However, participants consumed significantly more after doing a cognitive task than those who did not. In addition to that, Siervo et al. [37] similarly concluded that post-meal hunger levels were also not different between video game playing participants who experienced more stress compared to the television viewing participants, although participants playing video games ate significantly more than those watching television. Contrary to findings in this study, it has been found that people who were more stressed would consume more food to reach similar satiety compared to those who were less stressed [14,15,37]. The reason for discrepancy in these findings compared to results in this study is unclear. Indeed, when attention is focused on concurrent work at a meal time, appetite sensation, and eating behavior tend to interplay in a complicated way [13]. 

### 4.6. Energy Intake Did Not Differ between Restrained and Non-Restrained Eaters

Dietary restrained eaters always tend to eat less to lose or maintain weight. However, the present study found no significant differences in energy intake between low and high restrained consumers. One possible explanation is that dietary restriction might only exist in overweight people because they are motivated to limit their food consumption for losing weight. However, normal weight people are also likely to have this behavior as they often concentrate on their body shape and appearance [18]. This finding was consistent with results reported by [21] who also found no significant differences in terms of restrained eating behavior (using DBEQ) and energy intake between low-stress (completing solvable anagrams) and high-stress (completing unsolvable anagrams) conditions while eating snacks. Furthermore, Salama et al. [18] also concluded that there was no interaction between restrained eating behavior and food intake in both rest and reading–writing conditions. Moreover, Rutters et al. [16] concluded that dietary restraint status was not correlated to food intake in both control (doing solvable mental arithmetic task) and stress (doing unsolvable mental arithmetic task) conditions. 

## 5. Conclusions

This study set out to assess the effects of performing mental tasks while working on concurrent meal intake and to determine if restrained eating behavior influenced energy intake under a simulated stressful work condition. The results of this investigation showed that eating while working on a computer affected satiety, through significantly lowering post-meal hunger scores compared to those who did nothing while eating. Hence, this research supports the idea that eating while working can affect people’s eating behavior. The contribution of this study has been to confirm that energy intake of normal weight participants was not significantly influenced by mental work when eating while working. Furthermore, results indicated that there was no association between individual restraint eating behavior and energy intake. Further work should be done to establish the long-term effects of eating while working by monitoring 24-hour food intake of participants who complete mental tasks rather than just focusing on the current meal intake. Furthermore, the impact of stress caused by working on a computer on food intake with obese participants needs to be further investigated. A limitation of this study was the failure to consider the effect of the follicular phase and the luteal phase of female participants on energy intake, since women may behave differently when eating with stress. Additionally, the reliance of this study on self-reported measures makes it vulnerable to response bias. As such, electrophysiological measures can be further used to validate mood/stress responses [44,45]. One of the strengths of this study is that it represents a comprehensive examination of how state anxiety, stress level, and individual eating trait profiles influenced energy intake and appetite rating. Furthermore, a repeated measure design was also implemented to limit noise from individuals’ variance. This particular research finding also points to the need for exploring the effect of mental tasks that induce high cognitive load and other environmental distractions concurrently on energy intake.

## Figures and Tables

**Figure 1 nutrients-11-01545-f001:**
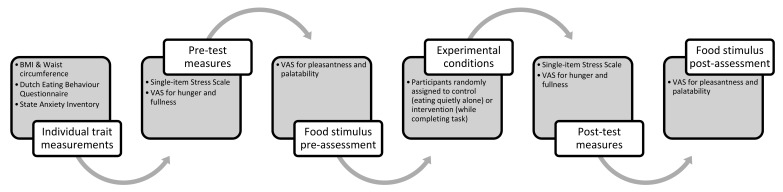
Overview of the experimental procedure.

**Figure 2 nutrients-11-01545-f002:**
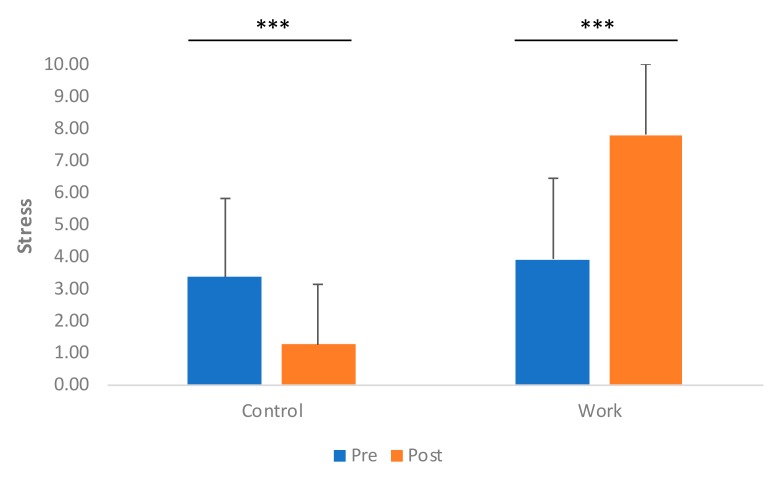
Self-rated stress for the control and work conditions. Pre- and post- significant differences were observed at the ***: 0.1% level. Error bars represent the standard deviation.

**Figure 3 nutrients-11-01545-f003:**
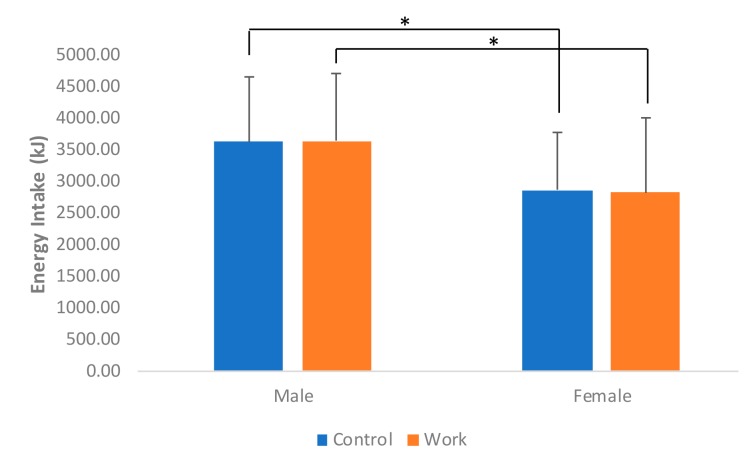
Normalized energy intake under control and work conditions. Significant difference was observed at the *: 5% level.

**Figure 4 nutrients-11-01545-f004:**
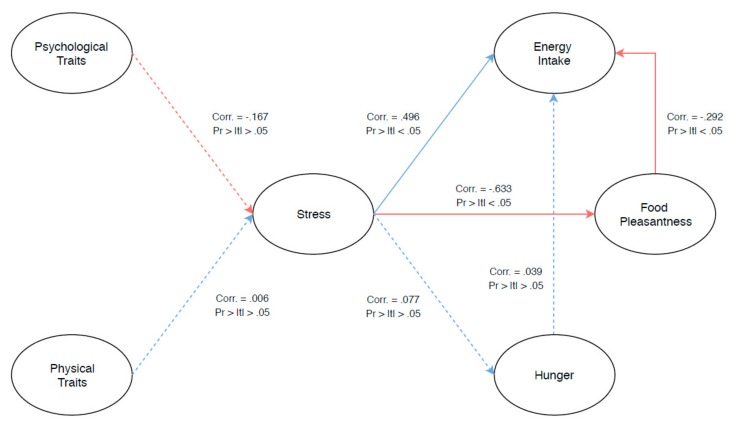
Partial least squares path modeling (PLS-PM) showing the relationship between physical (BMI, Waist circumference, Age, and Gender) and psychological measures (self-rated stress, State–Trait Anxiety Inventory (STAI), and Dutch Eating Behavior Questionnaire (DEBQ) dimensions), as well as energy intake, hunger, and food pleasantness. Solid and dotted lines correspond to significant (*p* < 0.05) and non-significant relationship (*p* > 0.05), respectively, between the latent variables.

**Table 1 nutrients-11-01545-t001:** Satiety visual analog scale.

Questions	Anchors	Reference
**How hungry are you right now?**	Not at all hungry and extremely hungry	[33]
**How full do you feel?**	Not at all full and totally full	[34]
**How strong is your desire to eat now?**	Very weak and very strong	[35]
**How much pizza do you think you could consume right now?**	Nothing at all and a large amount	[33]
**How satiated do you feel?**	I am completely empty and I cannot eat another bite	[34]

**Table 2 nutrients-11-01545-t002:** Food pleasantness visual analog scale.

Questions	Anchors	Reference
**How pleasant is the appearance of the pizza right now?**	Not at all pleasant and extremely pleasant	[33]
**How pleasant is the odor of the pizza right now?**		
**How pleasant did you find the pizza?**		
**How pleasant do you think this pizza would taste right now?**		
**How much do you think you would enjoy eating this pizza?**	Not at all and extremely	

**Table 3 nutrients-11-01545-t003:** Participant characteristics (results expressed as mean ± SD, and range (min and max) indicated in parentheses).

	Total (*n* = 43)	Male (*n* = 14)	Female (*n* = 29)
**Age (years)**	25.79 ± 4.87 (19–24)	25.29 ± 4.43 (20–38)	26.03 ± 5.12 (19–41)
**BMI (kg/m^2^)**	21.75 ± 2.75 (16.49–27.78)	22.28 ± 2.59 (16.98–26.31)	21.50 ± 2.83 (16.49–27.78)
**Waist (cm)**	76.08 ± 7.25 (62–90)	79.79 ± 6.72 (68–90)	74.29 ± 6.90 (62–90)
**DEBQ** **restraint eating**	2.46 ± 0.77 (1–4)	2.34 ± 0.90 (1–3.8)	2.51 ± 0.71 (1.1–4)
**DEBQ** **emotional eating**	2.49 ± 1.04 (1–4.92)	2.36 ± 1.26 (1–4.77)	2.56 ± 0.94 (1–4.92)
**DEBQ** **external eating**	3.28 ± 0.49 (1.9–4.3)	3.38 ± 0.43 (2.7–4.3)	3.23 ± 0.52 (1.9–4.1)
**STAI**	26.09 ± 8.53 (12–46)	27.5 ± 9.31 (13–46)	25.41 ± 8.13 (12–43)

DEBQ Dutch Eating Behavior Questionnaire [23]; STAI State–Trait Anxiety Inventory [24].

**Table 4 nutrients-11-01545-t004:** Descriptive results on outcome variables (results expressed as mean ± SD).

	Control	Work
**Stress**	1.26 ± 1.85	7.79 ± 2.2
**Energy Intake (kJ)**	3105.17 ± 1101.32	3083.87 ± 1162.03
**Hunger**	13.79 ± 4.42	19.3 ± 2.22
**Food Pleasantness**	4.47 ± 1.29	3.69 ± 1.27

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
