# Peer review of "How is Satiety Affected When Consuming Food While Working on A Computer?"

_nutrients, 2019, doi:10.3390/nu11071545_

Round 1
Reviewer 1 Report
This study was to examine the impact of any work-related stress on energy intake and appetite. It was a very interesting paper since most working adults often consume their lunch at their work desk in our modern society as a natural phenomenon.
The abstract was too long introduction and methods with too short results.
Line 24-25, “post-eating satiety level was significantly higher in the work compared to the control condition.” Please specify findings in more detail.
Intro: Line 57-59, not smoothly linked to the previous sentence. Please re-write this sentence in order for readers to know why this sentence was added.
Methods: It is highly recommended to register any human being study on clinical trial primary registry (i.e. CTgov or any relevant registry). Did authors register their clinical trial on any registry?
Data analysis description was not detail enough overall- see below.
Please justify this sample size; please describe if/how power analysis was conducted (a target main outcome and mean and standard deviation with the reference) including an attrition rate.
In Figure 3, please explain what “Normalized” means in the statistic section and why used normalized data for Energy Intake.
When performing statistical analyses, how was baseline treated?
For gender differences, were women studied during follicular phase vs Luteal phase? How can it influence study results in women on satiety and stress?
Regarding “Ensuring that participants’ appetite was at the same level between the work and control conditions before food consumption is necessary in order to measure subsequent energy intake and satiety after consuming food.”, how was it ensured? Which appetite levels?
Regarding “the present study found no significant differences in energy intake between low and high restrained consumers.” how was it categorized?
Please discuss about the study weakness.
A Take-home message from this study should be included. What was a detailed suggestion for further research?
Author Response
Reviewer 1
This study was to examine the impact of any work-related stress on energy intake and appetite. It was a very interesting paper since most working adults often consume their lunch at their work desk in our modern society as a natural phenomenon.
The abstract was too long introduction and methods with too short results.
RESPONSE: We have reduced the abstract in size while also adding further detail of the results:
The cut sentences: “There has been substantial research undertaken to investigate the influence of distraction and stress on people’s eating behaviour.” “Restrained eating behaviour was determined using The Dutch Eating Behavior Questionnaire (DEBQ).”
The amended sentence: “Measurements included BMI, energy intake, state anxiety, restrained eating behaviour, stress levels (pre- and post-eating), and appetite (before and after both work and control sessions).”
Line 24-25, “post-eating satiety level was significantly higher in the work compared to the control condition.” Please specify findings in more detail.
RESPONSE: The sentence now reads: “post-eating hunger levels were significantly higher in the work condition than the control condition.”
Intro: Line 57-59, not smoothly linked to the previous sentence. Please re-write this sentence in order for readers to know why this sentence was added.
RESPONSE: The sentence now reads: “Besides the different effects of various environmental stimulus on appetite ratings, the biological demands of mental work differ from physical work.”
Methods: It is highly recommended to register any human being study on clinical trial primary registry (i.e. CTgov or any relevant registry). Did authors register their clinical trial on any registry?
RESPONSE: It would probably be a step too far to classify our research as a clinical trial as such. However, this study was approved by the Auckland University of Technology Ethics Committee (AUTEC) with ethical approval (EA 18/179).
Data analysis description was not detail enough overall - see below.
Please justify this sample size; please describe if/how power analysis was conducted (a target main outcome and mean and standard deviation with the reference) including an attrition rate.
RESPONSE: A previous pilot study was conducted to gather some preliminary data, and it was determined that the Cohen’s d, which was approx. 0.8 was sufficient to achieve a statistical power of 0.85 (n = 38) to 0.90 (n = 44) with 40 participants. A statement on the power has been added in Section 2.2. As recruitment was targeted upon individuals who ate pizza, and pizza is generally a liked food by most, no attrition was anticipated nor experienced.
In Figure 3, please explain what “Normalized” means in the statistic section and why used normalized data for Energy Intake.
RESPONSE: Energy intake (kcal/kJ) was calculated and translated based on the weight of pizza consumed (g). We have clarified this in section 2.5.6.
When performing statistical analyses, how was baseline treated?
RESPONSE: The data from the two conditions are treated the same.
For gender differences, were women studied during follicular phase vs Luteal phase? How can it influence study results in women on satiety and stress?
RESPONSE: Thanks, this variable was not consider a priori. This is one of weakness of this study, potentially leading to a Type II error in terms of a gender effect. We have added this limitation to our conclusion.
Regarding “Ensuring that participants’ appetite was at the same level between the work and control conditions before food consumption is necessary in order to measure subsequent energy intake and satiety after consuming food.”, how was it ensured? Which appetite levels?
RESPONSE: Self-reported hunger levels did not differ significantly between the two groups before eating pizza. We have clarified the matter in the first two sentences in the Section 4.5.
Regarding “the present study found no significant differences in energy intake between low and high restrained consumers.” how was it categorized?
RESPONSE: Using the Dutch Eating Behaviour Questionnaire. Participants with restrained scores greater-or-equal to three were high restrained eaters, while those with restrained scores less than three were low restricted eaters. We apologise for this omission and have documented this criterion in the first two sentences in the Section 2.5.2.
Please discuss about the study weakness.
RESPONSE: We have added the weakness in the Section 5. “However, this study failed to consider the effect of follicular phase and luteal phase of female participants on energy intake since women may behave differently when eating with stress.”
A Take-home message from this study should be included. What was a detailed suggestion for further research?
RESPONSE: We have described ideas for future research in the manuscript’s conclusion.
Reviewer 2 Report
Ding et. al. present a very interesting study aiming to determine whether eating while working affects energy intake, stress level and satiety. The association between dietary restraint and the other variables is also studied. The research questions are relevant and methods are clearly presented. However, I have several concerns related mostly to the presentation of the results, which could be done more precisely. See below my comments and suggestions.
Major comments
Point 1: Even though the idea of the study becomes clear from the Introduction, current description of the objectives (lines 82-84) is vague. The objectives of the present study should be more precisely described. Please present the specific objectives or research questions which this study aims to answer.
Point 2: Results are not clearly presented.
First, results should be linked to the objectives specified in the introduction. Now there are results from several analyses presented, but the rationale behind all analyses is not described in introduction. For example, comparing energy intakes between genders. Please include all your main analyses in your aims, otherwise make clear which are just supplementary analyses.
Second, throughout the Results section authors report only the significance of the finding but the difference/effect should also be quantified. Please present also the descriptive results, e.g. means and standard deviations or effect sizes for all analyses you have made. I suggest adding a table or two. For example, “for energy intake, a significant effect was observed for age (F(10,85) = 6.89; p < .05)” does not tell how age is associated to energy intake. Do younger eat more and older less, or vice versa? Chek this throughout the whole Results section.
Third, if p-value is greater than 0.001, I suggest to present the exact value. That leaves the reader better equipped to make his/her own decisions about the significance of the finding. Please check this throughout the manuscript including the figures.
Fourth, it is not always clear when authors are referring to baseline or post-eating values of a certain measure. Please make sure that this is clear throughout the paper.
Point 3: Authors did not find difference between energy intake in control or eating while working conditions, even though they expected that work might distract an individual and lead to higher energy intake. However, they did find that perceived hunger decreased less in work condition compared to control despite of the similar energy intake. This is a very relevant finding, since it may lead to increased energy intake later during that day, even though it could not be measured in this study. This may have an impact on long-term weight management. The finding is nicely discussed in the Discussion section (paragraph 4.5), but I suggest that it should also be presented more thoroughly already in the Result section, including descriptive results as I commented earlier. That is, how big the difference actually was in addition to being statistically significant.
Point 4: I suggest you to include the most relevant numerical results in the Abstract. Furthermore, a sentence “However, post-eating satiety level was significantly higher in the work compared to the control condition.” sounds like the participants were more satiated, that is less hunger after work condition. Isn’t this on the contrary what you found?
Minor comments
Point 5: Line 34-35, reference for prevalence of obesity is missing.
Point 6: This is just a stylistic suggestion, but you do not need to write down all authors when referencing. If there are more than two authors you may use only the name of the first author followed by et al. You may also consider, could you reference to the previous studies without mentioning the names of the authors. It could improve the readability.
Point 7: Line 71-72, Dietary restraint is generally not considered a synonym for chronic dieting. However, it may depend on the definition of chronic dieting which is not exact. If you wish to consider dietary restraint as a synonym for chronic dieting, I suggest you to define what you mean by chronic dieting in this paper.
Point 8: Line 92-93, “who have medical conditions” is too vaguely phrased. Please include more detailed information.
Point 9: Line 117-118, Being restrained, emotional or external eater is not an either-or option. For example, one can have high scores in all these dimensions.
Point 10: Paragraph 2.5.1, Please include details about the anthropometrics. Add type and manufacturer of the scale and stadiometer used, as well as the details of the measurement. For example, was the height measured without shoes, how much clothing did participant have when the weight was measured.
Point 11: Paragraph 3.1 “Interestingly, further analysis showed…” Did you not except it to be so. I would argue that quietly eating pizza is likely to be relaxing… “These findings indicate that the assigned task in the work condition was sufficiently demanding.” I would leave this to the discussion.
Point 12: Paragraph 3.2, You present results comparing genders as well as effect of the age on energy intake. These were not included in you aims. It is not very surprising nor interesting that men eat more compared to women. I suggest you to replace the figure 3 with either a figure or a table related to your main research questions.
Point 13: Paragraph 3.2, “ANCOVA revealed significant differences for energy intake (F(10,85) = 2.78; p < .01) and hunger (F(10,85) = 6.25; p < .01) across control and work conditions, but not for food pleasantness ratings (F(10,85) = 1.63; p = .11).” I have to admit that I do not understand this sentence. If across control and work conditions means that they have been studied as a pooled sample, what is then compared. Is this total effect for all variables included in the model? It should be more clearly presented, what was included in the model.
Point 14: Paragraph 3.3, Please report the exact fit indices in addition to their relation to cut-off point.
Point 15: Paragraph 4.1, Your heading “Eating while working increased stress” may be misleading. Increased stress is likely to be result from working as you also mention in your discussion. Your study can not answer the question, whether eating while working increases stress more than just working. I suggest you modify the heading accordingly.
Point 16: Paragraph 4.4, You found that men ate more than women. Psychosocial mechanisms may apply explaining the finding, but could it be just that men are bigger with higher energy requirements and subsequently higher appetite. It is very usual finding in all studies that men consume more energy.
Point 17: The Discussion section lacks discussion of the strengths and weaknesses of the study as well as considerations about the generability of the results. What you consider as your biggest strengths compared to previous studies? Are there any weaknesses, for example how reliable are the measures you used?
Author Response
Reviewer 2
Ding et. al. present a very interesting study aiming to determine whether eating while working affects energy intake, stress level and satiety. The association between dietary restraint and the other variables is also studied. The research questions are relevant and methods are clearly presented. However, I have several concerns related mostly to the presentation of the results, which could be done more precisely. See below my comments and suggestions.
Major comments
Point 1: Even though the idea of the study becomes clear from the Introduction, current description of the objectives (lines 82-84) is vague. The objectives of the present study should be more precisely described. Please present the specific objectives or research questions which this study aims to answer.
RESPONSE: We have re-expressed the aims. The sentences now read: “The aim of this research was to determine if a work-related stressor can influence satiety. In two experimental conditions, one with and one without a work task, measures of hunger, fullness and food pleasantness will be compared. In addition, the influence of dietary restraint upon eating while working in terms of energy intake will be investigated.”
Point 2: Results are not clearly presented.
First, results should be linked to the objectives specified in the introduction. Now there are results from several analyses presented, but the rationale behind all analyses is not described in introduction. For example, comparing energy intakes between genders. Please include all your main analyses in your aims, otherwise make clear which are just supplementary analyses.
RESPONSE: Comparing energy intake between genders is a supplementary analysis. Salama, et al. compared the difference before, and this study sought to replicate their findings but, ultimately, found a different result. The difference has been discussed in the section 4.4.
Second, throughout the Results section authors report only the significance of the finding but the difference/effect should also be quantified. Please present also the descriptive results, e.g. means and standard deviations or effect sizes for all analyses you have made. I suggest adding a table or two. For example, “for energy intake, a significant effect was observed for age (F(10,85) = 6.89; p < .05)” does not tell how age is associated to energy intake. Do younger eat more and older less, or vice versa? Check this throughout the whole Results section.
RESPONSE: We agree with the Reviewer that the direction and magnitude of effects should be included., Hence we have added a t-value alongside with the F-values for the ANCOVA analysis to infer the direction of the effect.
Third, if p-value is greater than 0.001, I suggest to present the exact value. That leaves the reader better equipped to make his/her own decisions about the significance of the finding. Please check this throughout the manuscript including the figures.
We have changed the p-values to the exact p-value especially when it is > 0.001.
Fourth, it is not always clear when authors are referring to baseline or post-eating values of a certain measure. Please make sure that this is clear throughout the paper.
RESPONSE: We have checked and clarified all the values.
Point 3: Authors did not find difference between energy intake in control or eating while working conditions, even though they expected that work might distract an individual and lead to higher energy intake. However, they did find that perceived hunger decreased less in work condition compared to control despite of the similar energy intake. This is a very relevant finding, since it may lead to increased energy intake later during that day, even though it could not be measured in this study. This may have an impact on long-term weight management. The finding is nicely discussed in the Discussion section (paragraph 4.5), but I suggest that it should also be presented more thoroughly already in the Result section, including descriptive results as I commented earlier. That is, how big the difference actually was in addition to being statistically significant.
RESPONSE: We agree with the reviewer on this point. We have added the mean and standard deviation for Hunger as directed by the Reviewer (re: Section 3.2)
Point 4: I suggest you to include the most relevant numerical results in the Abstract. Furthermore, a sentence “However, post-eating satiety level was significantly higher in the work compared to the control condition.” sounds like the participants were more satiated, that is less hunger after work condition. Isn’t this on the contrary what you found?
RESPONSE: The sentence now reads: “However, post-eating hunger level was significantly higher in the work compared to the control condition.”
Minor comments
Point 5: Line 34-35, reference for prevalence of obesity is missing.
RESPONSE: Missing reference added.
Point 6: This is just a stylistic suggestion, but you do not need to write down all authors when referencing. If there are more than two authors you may use only the name of the first author followed by et al. You may also consider, could you reference to the previous studies without mentioning the names of the authors. It could improve the readability.
RESPONSE: Thank you, and we agree. Changes made accordingly.
Point 7: Line 71-72, Dietary restraint is generally not considered a synonym for chronic dieting. However, it may depend on the definition of chronic dieting which is not exact. If you wish to consider dietary restraint as a synonym for chronic dieting, I suggest you to define what you mean by chronic dieting in this paper.
RESPONSE: It now reads: “and is a synonym for chronic dieting, which is a state of trying to restrict food intake.”
Point 8: Line 92-93, “who have medical conditions” is too vaguely phrased. Please include more detailed information.
RESPONSE: Phrase now reads: “….or who have medical conditions or take medication which can affect appetite”
Point 9: Line 117-118, Being restrained, emotional or external eater is not an either-or option. For example, one can have high scores in all these dimensions.
RESPONSE: The sentence now reads: “…if they were restrained, emotional and external eaters.”
Point 10: Paragraph 2.5.1, Please include details about the anthropometrics. Add type and manufacturer of the scale and stadiometer used, as well as the details of the measurement. For example, was the height measured without shoes, how much clothing did participant have when the weight was measured.
RESPONSE: We now have included details about the anthropometrics. “Height and waist circumference were measured by a flexible, but non-stretchable measuring tape. Weight was measured by a scale placed on the ground. Participants were asked to take off their heavy outerwear such as jackets and coats, and shoes.”
Point 11: Paragraph 3.1 “Interestingly, further analysis showed…” Did you not except it to be so. I would argue that quietly eating pizza is likely to be relaxing… “These findings indicate that the assigned task in the work condition was sufficiently demanding.” I would leave this to the discussion.
RESPONSE: Many previous studies just measured stress before, but not after, eating. This study measured stress level before and after eating pizza to make the result more valid. The sentence now reads: “Further analysis showed that the post-control condition showed significantly lower stress ratings compared to the pre-control condition (F(1,85) = 19.78; p < .01).”
Point 12: Paragraph 3.2, You present results comparing genders as well as effect of the age on energy intake. These were not included in you aims. It is not very surprising nor interesting that men eat more compared to women. I suggest you to replace the figure 3 with either a figure or a table related to your main research questions.
RESPONSE: Comparing energy intake between genders as well as across age were supplementary analyses. This is just to confirm the results of previous studies, but our findings do not concur with these previous results, which we explore in the discussion.
Point 13: Paragraph 3.2, “ANCOVA revealed significant differences for energy intake (F(10,85) = 2.78; p < .01) and hunger (F(10,85) = 6.25; p < .01) across control and work conditions, but not for food pleasantness ratings (F(10,85) = 1.63; p = .11).” I have to admit that I do not understand this sentence. If across control and work conditions means that they have been studied as a pooled sample, what is then compared. Is this total effect for all variables included in the model? It should be more clearly presented, what was included in the model.
RESPONSE: This means that EI and hunger ratings reached significance, but food pleasantness ratings did not. In the ANCOVA model we have included the physiological (e.g. BMI, Waist circumference, Age) and psychological (DEBQ, SAI) parameters as covariates. The main effect analyses were then carried out for gender and the experimental condition (work vs control). This has been described more clearly in Section 2.6 which reads
“Analysis of covariance (ANCOVA) was carried out to explore how the dependent variables of energy intake, hunger ratings, and food pleasantness were influenced in the control and work conditions, while statistically controlling for the effects of other continuous variables, which are not of primary interest like physical characteristics (BMI, waist circumference, age, and gender) and psychological traits (DEBQ & STAI scores) that are known as covariates.”
Point 14: Paragraph 3.3, Please report the exact fit indices in addition to their relation to cut-off point.
RESPONSE: The exact value has been added to the manuscript. Which now reads:
“Cronbach's α, Dillon-Goldstein's rho, and R2 values of each variable were 0.822, 0.854, and 0.836 respectively, indicating overall reliability of the measured variables.”
Point 15: Paragraph 4.1, Your heading “Eating while working increased stress” may be misleading. Increased stress is likely to be result from working as you also mention in your discussion. Your study can not answer the question, whether eating while working increases stress more than just working. I suggest you modify the heading accordingly.
RESPONSE: We have changed the heading into “Eating while working influenced stress.”
Point 16: Paragraph 4.4, You found that men ate more than women. Psychosocial mechanisms may apply explaining the finding, but could it be just that men are bigger with higher energy requirements and subsequently higher appetite. It is very usual finding in all studies that men consume more energy.
RESPONSE: We agree with the Reviewer’s assumption, but interestingly it is not a given that men eat more than women in studies such as this, and we discuss this in in paragraph 4.4.
Point 17: The Discussion section lacks discussion of the strengths and weaknesses of the study as well as considerations about the generability of the results. What you consider as your biggest strengths compared to previous studies? Are there any weaknesses, for example how reliable are the measures you used?
RESPONSE: We have added: “One of the strengths of this study is that it represents a comprehensive examination of how state anxiety, stress level, and individual eating trait profile influenced energy intake and appetite rating. Furthermore, a repeated measure design was also implemented to limit noise from individuals’ variance.”
We have also added a sentence in the Section 5. “However, this study failed to consider the effect of follicular phase and luteal phase of female participants on energy intake since women may behave differently when eating with stress. Additionally, reliance of this study on self-reported measures make it vulnerable to response bias.”
Round 2
Reviewer 2 Report
Authors have done comprehensive work answering my questions and concerns. However, I still have a couple of suggestions.
My first comment is related to my initial point 2 about clearness of the results. More specifically, I would still like to recommend authors to add a table including descriptive results (mean and standard deviation) of outcome variables (energy intake, hunger, stress, and food pleasantness) before and after eating in both conditions. Even though the results section now includes the relevant information, it is very heavy to read. If a table was included, it would be easier to get a comprehensive picture. But I leave this up to authors.
My second comment is related to my initial point 12 about comparing genders. After authors response, I have now a better idea, why this analysis was done. The reason was not only to compare energy intake between genders, but to determine whether men and women respond similarly to different eating conditions. I think the Figure 3 was a bit misleading, because currently it looks like it mainly compares genders. If I understood right, the main aim was to compare two conditions within genders. I suggest to change the layout of the figure so that the color of the bars refers to condition, and different conditions are presented next to each other for both genders. This way the main message - energy intake is not affected by condition neither in men nor women - comes more clear.
Author Response
My first comment is related to my initial point 2 about clearness of the results. More specifically, I would still like to recommend authors to add a table including descriptive results (mean and standard deviation) of outcome variables (energy intake, hunger, stress, and food pleasantness) before and after eating in both conditions. Even though the results section now includes the relevant information, it is very heavy to read. If a table was included, it would be easier to get a comprehensive picture. But I leave this up to authors.
RESPONSE: We have added Table 4 to provide an overview of the descriptive results of this study.
My second comment is related to my initial point 12 about comparing genders. After authors response, I have now a better idea, why this analysis was done. The reason was not only to compare energy intake between genders, but to determine whether men and women respond similarly to different eating conditions. I think the Figure 3 was a bit misleading, because currently it looks like it mainly compares genders. If I understood right, the main aim was to compare two conditions within genders. I suggest to change the layout of the figure so that the color of the bars refers to condition, and different conditions are presented next to each other for both genders. This way the main message - energy intake is not affected by condition neither in men nor women - comes more clear.
RESPONSE: We have revised Figure 3 by shuffling the colour coding according to the reviewer’s directives.
We would like to thank the Reviewer for their helpful and constructive comments and for the opportunity to submit our revisions.